# Clinical Significance of Analysis of Vitamin D Status in Various Diseases

**DOI:** 10.3390/nu12092788

**Published:** 2020-09-11

**Authors:** Magdalena Kowalówka, Anna K. Główka, Marta Karaźniewicz-Łada, Grzegorz Kosewski

**Affiliations:** 1Department of Bromatology, Poznan University of Medical Sciences, 42 Marcelińska Street, 60-354 Poznań, Poland; mkowalowka@ump.edu.pl (M.K.); aglowka@ump.edu.pl (A.K.G.); grzegorzkosewski@ump.edu.pl (G.K.); 2Department of Physical Pharmacy and Pharmacokinetics, Poznan University of Medical Sciences, 6 Święcickiego Street, 60-781 Poznań, Poland

**Keywords:** calcidiol, calcitriol, liquid chromatography coupled with tandem mass spectrometry (LC-MS/MS), vitamin D receptor, vitamin D status

## Abstract

Vitamin D plays a role not only in the proper functioning of the skeletal system and the calcium-phosphate equilibrium, but also in the immune system, the cardiovascular system and the growth and division of cells. Although numerous studies have reported on the analysis of vitamin D status in various groups of patients, the clinical significance of measurements of vitamin D forms and metabolites remains ambiguous. This article reviews the reports analyzing the status of vitamin D in various chronic states. Particular attention is given to factors affecting measurement of vitamin D forms and metabolites. Relevant papers published during recent years were identified by an extensive PubMed search using appropriate keywords. Measurement of vitamin D status proved to be a useful tool in diagnosis and progression of metabolic syndrome, neurological disorders and cancer. High performance liquid chromatography coupled with tandem mass spectrometry has become the preferred method for analyzing the various forms and metabolites of vitamin D in biological fluids. Factors influencing vitamin D concentration, including socio-demographic and biochemical factors as well as the genetic polymorphism of the vitamin D receptor, along with vitamin D transporters and enzymes participating in vitamin D metabolism should be considered as potential confounders of the interpretation of plasma total 25(OH)D concentrations.

## 1. Introduction

Vitamin D involves six different steroid hormones with different activities: the inactive endogenous precursor—cholecalciferol (D_3_), partially active calcidiol (25(OH)D_3_), its active dihydroxy-form calcitriol (1,25(OH)_2_D_3_), a plant-derived inactive form—ergocalciferol (D_2_), and its metabolites, 25(OH)D_2_ and 1,25(OH)_2_D_2_ [1]. Calcitriol has a key role to play in calcium and phosphate homeostasis, cell proliferation and differentiation, and in the responses of the immune and nervous systems [2]. In the last two decades novel functions of vitamin D have been discovered including protection against cardiovascular disease, diabetes and colorectal cancer, neuroprotective activity, oxidative stress reduction, xenobiotic detoxification, and antimicrobial and anti-inflammatory activity [3]. Vitamin D’s mechanism involves binding calcitriol to the vitamin D receptor (VDR), which is a transcription factor positively or negatively regulating the expression of genes that mediate its biological activity. Wide distribution of the VDR in all cell types may explain its multiple actions on different tissues [4]. As the function of VDR may be influenced by circulating levels of vitamin D, assessment of vitamin D status in the body may be clinically valuable to predict patients’ susceptibility to certain diseases. Only 25-hydroxy vitamin D and 1,25-dihydroxyvitamin D have clinical significance, thus total 25-hydroxy vitamin D concentration is the most valuable for evaluating vitamin D status in the body and for subsequent therapeutic decisions [5]. There are many commercially available 25-hydroxyvitamin D methods based on the immunoassay technique used by laboratories for determination of vitamin D status. Recently, high performance liquid chromatography coupled with tandem mass spectrometry (HPLC-MS/MS) has become the preferable method because it overcomes inaccuracy problems associated with immunoassays and protein binding assays [6]. The underlying causes of inter-assay variability involve tight binding of vitamin D metabolites to vitamin D-binding protein, cross-reaction with 24,25-dihydroxyvitamin D, or various specificities against 25(OH)D_2_ and 25(OH)D_3_. Moreover, the HPLC-MS/MS technique allows the separation of C3-epimers of vitamin D metabolites, differing from the primary molecules only in stereochemistry. Lack of such separation may lead to inaccuracies in 25(OH)D measurements [7].

A number of environmental and biological factors influence vitamin D status. These factors include variation in sun exposure, age, sex, obesity, or chronic illnesses [8]. Moreover, it has been shown that polymorphisms of genes involved in the metabolism, catabolism, transport, or binding of vitamin D to VDR might affect vitamin D concentrations [9].

This review evaluates the role of vitamin D measurements in prediction and treatment of certain diseases, summarizes analytical techniques used for determination of vitamin D concentrations, and briefly discusses the contribution of various factors to vitamin D status in specific populations.

## 2. Synthesis and Metabolism

The main sources of vitamin D are certain foods, dietary supplements, and exposure to ultraviolet B (UVB) radiation from sunlight (290–315 nm) [10]. Diet sources include oily sea fish and fish oil, and to a lesser extent egg yolks and dairy products, mainly in the form of vitamin D_3_ (cholecalciferol), which is highly bioavailable. Vitamin D_2_ (calciferol) is present in products of plant origin and in mushrooms. However, most diets contain only small amounts of vitamin D, and in some countries few foods, such as milk or margarine, are fortified with vitamin D [10,11]. Ninety percent of the body’s pool of this compound constitutes skin synthesis. Endogenous vitamin D_3_ synthesis occurs in the keratinocytes. Stimulation of 7-dehydrocholesterol by UVB leads to pre-vitamin D_3_ formation, and depends on skin pigmentation and on the latitude, season, and time of day [10]. Then pre-vitamin D_3_ undergoes thermal isomerization to form vitamin D_3_ or converts to lumisterol and tachysterol under UVB irradiation [12].

Vitamin D_2_ and D_3_ are biologically inactive and their serum concentration is very low. According to the literature data, the concentration of D_3_ in normal subjects was 6.6 ± 3.0 ng/mL while D_2_ was below limit of detection [13]. Both vitamins, D_2_ and D_3_, are highly lipophilic and must be carried in the aqueous environment of blood by protein transporters. Vitamin D_2_ and D_3_ taken orally are absorbed from the intestines, and then transported by chylomicrons, which are further processed into lipoproteins. Endogenously produced D_3_ is carried by D-binding protein (DBP) [1]. In the liver, the hydroxylation of both vitamin D_2_ and D_3_ leads to the formation of the biologically inactive prohormones 25(OH)D_2_ and 25(OH)D_3_ (calcifediol). This process is catalyzed by enzymes possessing 25-hydroxylase activity, such as CYP2R1 and CYP27A1 [4] (Figure 1). Currently, the total serum concentration of 25(OH)D is considered the best indicator of vitamin D status in the body [14]. The Institute of Medicine suggests that serum 25(OH)D concentrations greater than 20 ng/mL (50 nmol/L) can be considered sufficient, while values between 12 and 20 ng/mL (30–50 nmol/L) are insufficient, and lower than 12 ng/mL (30 nmol/L) are deficient [15]. The dominating 25(OH)D form circulating in the body is 25(OH)D_3_ with concentration in the range of 2.2–42.7 ng/mL in healthy subjects, while the level of 25(OH)D_2_ is 0.2–15.9 ng/mL [16]. Further hydroxylation of 25-hydroxyvitamins takes place mainly in the kidney by 1*α*-hydroxylase (mitochondrial CYP27B1) resulting in production of 1,25(OH)_2_D_2_ and 1,25(OH)_2_D_3_ (calcitriol). This process is regulated by serum calcium and phosphate, as well as parathyroid hormone (PTH) [17]. 1,25(OH)_2_D is the most biologically active form of vitamin D. It controls intestinal absorption of calcium and phosphate, stimulates osteoclast activity, and helps regulate the release of parathyroid hormone [2]. Both forms 1,25(OH)_2_D_2_ and 1,25(OH)_2_D_3_ can bind to VDR, although 1,25(OH)_2_D_3_ has proven to be significantly more potent [18]. Reference values determined in healthy subjects by Dirks et al. were 59–159 pmol/L for 1,25(OH)_2_D_3_ and <17 pmol/L for 1,25(OH)_2_D_2_ [19].

25(OH)D and 1,25(OH)_2_D can be converted by CYP24A1 to 24,25(OH)_2_D and 1,24,25(OH)_3_D, respectively. 24,25(OH)_2_D appears to be required for optimal endochondral bone formation and bone fraction repair [20]. It possesses a circulating half-life of seven days and concentrations in ng/mL (0.084–9.514 ng/mL [16]). The metabolite is regarded as a useful marker of vitamin D catabolism [21]. In patients with CYP24A1 mutations, 24,25(OH)_2_D levels are low or even undetectable despite the presence of adequate amounts of the substrate 25(OH)D. Low concentration of 24,25(OH)_2_D and normal-to-high concentrations of 25(OH)D have been found in children with idiopathic infantile hypercalcemia carrying mutations of CYP24A1 [22]. It was suggested that 1,24,25(OH)_3_D may have biological activity because it shows substantial affinity for the VDR [2]. The enzyme can also catalyze the C23 lactone pathway resulting in the formation of 1,25(OH)_2_D-26,23 lactone and 25(OH)D-26,23 lactone [4]. Thus, the primary function of CYP24A1 is to prevent the accumulation of 1,25(OH)2D and 25(OH)D. Moreover, vitamin D metabolites may undergo an epimerization process by 3-epimerase. The enzyme isomerizes the C-3 hydroxy group of the A ring from the α to β orientation. Clinical significance of the 3-epimers is unknown but reduced binding of 3-epi-25(OH)D to DBP and reduced affinity of 3-epi-1,25(OH)2D for the VDR were reported [2]. Evidence suggests that 3-epi-1,25(OH)_2_D suppresses PTH as effectively as 1,25(OH)_2_D. Moreover, 3-epi-1,25(OH)_2_D_3_ can boost the synthesis of surfactant phospholipids and affect gene expression to activate the synthesis of surfactant protein-B in pulmonary alveolar type II [23]. Levels of the 3-epimers in the human body are low and rarely detected. Serum concentrations of 3-epi-25(OH)D_3_ in normal subjects were in the range of 0.4–6.0 ng/mL [16]. In keratinocytes, 20-hydroxylation of vitamin D by CYP11A1 takes place, resulting in formation of 20(OH)D and 20,23(OH)2D. Biological activity of 20(OH)D_3_ was confirmed in the induction of keratinocyte differentiation [24]. It was reported that the compound inhibits DNA synthesis in epidermal keratinocytes, melanocytes, and melanoma cells through activation of VDR [25].

## 3. Mechanism of Action

1,25(OH)_2_D_3_, the active form of vitamin D, acts as steroid hormone by binding to VDR that is omnipresent throughout the body. VDR is a transcription factor that forms a complex with another intracellular receptor, the retinoid-X receptor (RXR). The genomic mechanism of vitamin D action involves the direct binding of 1,25(OH)_2_D_3_ activated VDR/RXR to specific DNA sequences called vitamin D response elements (VDREs) resulting in either activation or inhibition of transcription. By turning genes on or off, this complex controls intestine calcium and phosphate absorption, renal calcium reabsorption and phosphate loss, and bone homeostasis. Other VDR dependent effects involve regulation of hormone secretion, cell differentiation and proliferation, and immune function. 1,25(OH)_2_D inhibits PTH production and secretion by PTH gene suppression, stimulates insulin secretion in pancreatic beta cells, and stimulates fibroblast growth factor (FGF) production in osteoblasts and osteocytes. It was reported that vitamin D slows down cancer cell growth by such mechanisms as stimulation of the expression of cell cycle inhibitors p21 and p27, activation of the expression of the cell adhesion molecule E-cadherin, inhibition of the transcriptional activity of b-catenin, and controlling angiogenesis [26]. The immunomodulatory effect of 1,25(OH)_2_D_3_ involves stimulation of antimicrobial peptides, cathelicidin and defensin *β*2, reducing T cell proliferation, modulating T cell differentiation, and inhibition of the maturation of dendritic cells important for antigen presentation [27].

1,25(OH)_2_D also has effects on selected cells that do not involve gene regulation and are mediated by a membrane receptor. This mechanism regulates calcium and chloride channel activity, activation and distribution of protein kinase C, and phospholipase C activity in osteoblasts, liver, muscle, and intestine cells [2].

## 4. Methods of Analysis of Vitamin D in Biological Fluids

Serum levels of the biologically active metabolite 1,25(OH)_2_D may be difficult to measure because half-life of this compound is about 4 h and its production is regulated by hormones and minerals [28]. 25(OH)D is employed as a biomarker for overall vitamin D status because its half-life is two–three weeks and its concentration accounts for both dietary vitamin D intake and endogenous synthesis of vitamin D in the skin [14]. Moreover, levels of the main circulating form, 25(OH)D_3_, have been correlated to the onset and progression of many diseases [29,30]. Therefore, precise and accurate methods for analysis of 25(OH)D are essential to distinguish insufficient levels of the compound.

Vitamin D and its metabolites are demanding analytes to determine, because they are very lipophilic, at very low concentrations, and also have a high affinity for proteins that bind to vitamins [1]. There are several FDA-approved commercial 25(OH)D tests for routine clinical laboratory measuring of vitamin D status based on radioimmunoassay (RIA), enzyme immunoassay (EIA), chemiluminescent immunoassay (CLIA), and competitive protein-binding assay (CPBA) techniques [6].

### 4.1. Immunoassay Techniques

A large number of immunoassay methods for measurement of the vitamin D forms exist. They differ in antibodies used and as a result some of them measure only one form, 25(OH)D_2_ or 25(OH)D_3_, while others measure total 25(OH)D levels [31]. Recently, a rapid immunoassay-based point-of-need diagnostic test for the assessment of 25(OH)D_3_ in finger-stick blood was developed. The assay was accompanied by a smartphone-assisted imaging device which allows for easy operation and access to the results [32]. Moreover, a chemiluminescent immunoassay method was developed for determination of the sum of 1,25(OH)_2_D_2_ and 1,25(OH)_2_D_3_. However, the obtained results exhibited significant deviation from the values determined by a reference HPLC-MS/MS method [33]. It was reported that immunoassay methods showed significant discrepancies in results, demonstrating positive bias, which results in overestimating vitamin D deficiency [34]. Moreover, Lee et al. [35] suggest that difference in cross-reactivities may be the main cause of discrepancy between immunoassays.

### 4.2. HPLC-MS/MS

Recently, HPLC-MS/MS has come to represent the gold standard for vitamin D plasma evaluation because it can discriminate between all forms and metabolites of vitamin D. Moreover, MS/MS detection allows the analysis of much lower concentrations of vitamin D and its metabolites than previously used HPLC-UV methods which enabled the determination of 25(OH)D_2_ and 25(OH)D_3_ only. Recently, interest in the other forms and metabolites of vitamin D has arisen. The measurement of 25(OH)D using standard chromatographic methods may result in overestimated values because of the presence of the C-3 epimers, which may constitute up to 50% of the 25(OH)D content in adults [36,37]. Clinically, this issue is particularly important in estimating vitamin D status in infants where levels of the epimeric form may be higher than concentrations of 25(OH)D [38].

There are several HPLC-MS/MS methods for analysis of vitamin D in human serum, plasma, dry blood spots and milk [16,19,39,40,41,42,43,44,45,46,47,48,49,50,51,52]. Moreover, application of supercritical fluid chromatography (SFC) with MS/MS detection was reported [13,53]. These methods allow the analysis of not only the main metabolite 25(OH)D but also substrates, D_2_ and D_3_, as well as other metabolites including epimers and dihydroxyl derivatives (Table 1). In the methods described, the chromatographic separation of vitamin D and its metabolites was usually performed on a C18 column. For more efficient separation of multiple isomeric compounds, a chiral column with cellulose selector was also used [16,53]. The mobile phases were composed of organic reagents such as methanol or acetonitrile with the addition of an aqueous solution of formic acid, ammonium formate, ammonium acetate or methylamine. For SFC, mixtures of CO_2_ with methanol and formic acid were used (Table 1).

In these methods, an effective release of 25(OH)D from the binding protein can be assured by a simple sample preparation procedure including protein precipitation or alkaline hydrolysis followed by liquid-liquid extraction (LLE) with hexane or ethyl acetate or solid-phase extraction (SPE). In one case, immunoextraction was used to purify the sample [19]. The various extraction methods have enabled effective isolation of analytes from biological material, as evidenced by the recovery value in the range of 50–113% [48,49]. Some procedures required further derivatization with 4-(4′-dimethylaminophenyl)-1,2,4-triazoline-3,5-dione (DAPTAD) or 4-phenyl-1,2,4-triazoline- 3,5-dione were (PTAD) to improve sensitivity of the method in case of 1α,25(OH)_2_D_3_ [53] or to generate a stable product ion under MS ionization for 25(OH)D_3_ glucuronide [48]

## 5. Analysis of Vitamin D in Various Diseases

In the case of vitamin D deficiency in the body, it was proved that the parathyroid glands produce too much PTH, which then contributes to an increase in resorption in bone tissue metabolites, leading to a decrease in mineral density, which leads to a general feeling of tiredness and weakness, and as a consequence osteomalacia and osteoporosis [54]. Therefore, vitamin D deficiency may be associated with an increased risk of fracture [55].

Vitamin D deficiency very often occurs in patients with autoimmune diseases such as rheumatoid arthritis (RA), lupus erythematosus, or a number of other rheumatological conditions [56]. Therefore, it seems that in these conditions measurements of vitamin D levels may be very important (Table 2). Serum concentrations of metabolites such as 3-epi-25(OH)D_3_ and synovial fluid metabolites, especially 1,25(OH)_2_D_3_, were found to be more closely related to the progression and course of RA [55]. Li et al. [57] reported that 3-epi-25(OH)D_3_ was significantly lower in serum of RA patients and in resolving reactive arthritis (ReA) patients relative to healthy controls. Moreover, the authors found differences in concentrations of various vitamin D metabolites between serum and synovial fluid [57]. In RA, active forms of vitamin D, in response to current inflammation, reduce the number of Th1 and Th17 lymphocytes and reduce the production of pro-inflammatory cytokines, which are responsible for severe RA symptoms [58]. In another study [59], the effects of vitamin D supplementation in RA patients have contributed to improving patient health. Disease activity expressed by the mean disease activity index using C-reactive protein in these patients showed statistically significant improvement after supplementation (*p* = 0.002). Serum vitamin D levels increased from 10.05 ± 5.18 to 57.21 ± 24.77 ng/mL (*p* < 0.001) during treatment [59].

Vitamin D deficiency has been associated with autoimmune thyroid disease (Hashimoto’s, Graves’ disease) and non-inflammatory bowel disease. Serum 25(OH)D levels were significantly lower in patients with hypothyroidism compared to the control group (t = −11.11; *p* = 0.000). As a result of poor absorption caused by inflammation of the intestines, the deficiency of vitamin D is worsened due to impaired food absorption [60]. Another example of the effects of vitamin D deficiency is psoriasis—a chronic inflammatory disease that increases the risk of cardiovascular disease. A proportional relationship was demonstrated between 1,25(OH)_2_D and risk factors for cardiovascular disease, such as the volume of visceral adipose tissue (β = −0.43, *p* = 0.026 and β = −0.26 *p* = 0.13), uptake of fluorodeoxyglucose (FDG) in vessels (β = −0.19, *p* = 0.01) and coronary plaque load (β = −0.18, *p* = 0.03), regardless of traditional risk factors. It can be concluded that the level of 1,25(OH)_2_D helps to better capture the cardiometabolic risk associated with vitamin D deficiency [61].

Evidence suggests that insulin resistance syndrome (metabolic syndrome) and vitamin D deficiency contribute greatly to the formation of cancer [62]. High insulin levels may cause obesity, hypertension, low HDL cholesterol, high triglycerides, and diabetes [63]. It is interesting that vitamin D deficiency in the body contributes to the deepening of insulin resistance, resulting in metabolic disorders (hyperglycemia, dyslipidemia) [64].

The presence of a vitamin D receptor in endothelial cells, smooth blood vessels and cardiomyocytes and the presence of 1α-hydroxylase in the heart confirm the effect of this compound on the circulatory system. Regulation of renin-angiotensin-aldosterone (RAA), and modulation of the inflammatory response and tissue calcification process are involved in the pathogenesis of cardiovascular diseases, which include myocardial infarction, atrial fibrillation, or heart failure [65]. As a result of vitamin D deficiency, an increase in the concentration of parathyroid hormone is observed, which leads to the development of hypertension, endothelium dysfunction and calcification of the aortic valve. Calcitriol, by reducing the level of free radicals, has anti-inflammatory effects, thereby affecting the stabilization of atherosclerotic plaques [66].

Among the many positive properties of 1,25(OH)_2_D, the anti-cancer effect should also be taken into account. This compound works by inhibiting proliferation, activating apoptosis, stimulating differentiation, and inhibiting angiogenesis [67]. Studies have shown a directly proportional relationship between the incidence of given types of cancers and the concentration of vitamin D in serum and latitude. People with vitamin levels higher than 50 mmol/L had a lower risk of developing prostate and colorectal cancer by as much as 30–50% [12,68]. While a link exists between vitamin D levels of <30 nmol/L in the blood and increased incidence of colorectal cancer (31%), in the case of breast and prostate cancer no associations were found [69,70]. In the study of Manson et al., after one year of supplementation with 2000 IU of vitamin D, average levels of 25-hydroxyvitamin D increased by 40%. However, vitamin D supplementation did not result in a lower incidence of invasive cancer [71].

Vitamin D deficiency has also been noted in patients with chronic obstructive pulmonary disease (COPD), obesity, high triglycerides, diabetes and its complications, metabolic syndrome, and multiple sclerosis. For diabetes, measuring 1,25(OH)_2_D_3_ serum may help to predict the severity of diabetic retinopathy [72]. In people with metabolic syndrome, low vitamin D levels were more common in people with a higher BMI [73,74]. Low levels of 25(OH)D in the blood may contribute to a higher risk of developing multiple sclerosis in newborns. This risk of developing the disease was the highest at 25(OH)D levels <20.7 nmol/L, while at ≥48.9 nmol/L, risk was lowest [75].

Another inflammatory disease in which vitamin D pathogenesis can be important is bronchiolitis in infants. This may suggest a negative correlation between low concentrations of 25(OH)D and IgE in the serum of small patients [76]. Infants who suffered from food allergies were also diagnosed with vitamin D deficiencies that increased the likelihood of persistent allergies (especially in people with the GG genotype in vitamin D-binding protein). Antenatal supplementation has been shown to reduce the risk of food allergies in particular in infants with GT/TT genotype [77].

Low serum vitamin D levels may be associated with the onset of psychiatric disorders such as depression and post-traumatic stress disorder (PTSD) [78,79,80]. It was found that vitamin D is involved in the processes of maturation and differentiation of the cells of the nervous system, by affecting the synthesis of neural and glial cell lines of growth factors. It also controls the synthesis of the neurotransmitters acetylcholine (ACh), dopamine (DA) and γ-aminobutyric acid (GABA) [81]. Changed expression of the vitamin D receptor gene has been shown to be associated with Alzheimer’s disease. Vitamin D may have neuroprotective properties, protecting against the harmful effects of glutamate and reducing inflammation induced by the deposition of β-amyloid in the form of plaques [81]. Interesting conclusions are provided by the study of Jhee et al. [78] regarding the occurrence of depression in people with chronic kidney disease. It was noted that vitamin D deficiency was an important independent predictor of depression, and the incidence was significantly higher in patients with chronic kidney disease than in the general population (14.3 vs. 11.1%, *p* = 0.03). In addition, the study by Imai et al. [79] showed that stronger symptoms of depression occurred in those with the lowest levels of vitamin D. Due to vitamin D deficiency, anatomical changes in the hippocampus have been observed and serotonin levels and calcium levels in neurons have been reduced, which also predisposes to depression [82]. Insufficient skin synthesis of vitamin D in the autumn-winter period leads to a higher probability of the appearance of depressive states, which disappear in the summer [83]. Terock et al. [80] observed that deficient 25(OH)D levels were positively associated with post-traumatic stress disorders (OR = 2.02; *p* = 0.028). Additionally, in carriers of the CC-genotype of rs4588, significantly higher 25(OH)D levels were observed (*p* < 0.001) along with lower odds for the disease (OR = 0.35; *p* = 0.023) compared to the AA-genotype. Carriers of the TT-allele of rs7041 showed lower 25(OH)D levels (*p* < 0.001) and increased odds for the disease (OR = 2.80; *p* = 0.015) compared to the GG-genotype [80].

Vitamin D deficiency is a risk factor for acute respiratory infections (ARI). Meta-analysis by Martineau et al. [84] confirmed that vitamin D supplementation protects against ARI. Recently, several studies have started to clarify whether supplementation with vitamin D in different dosages has an influence on the course of the COVID-19 disease caused by severe acute respiratory syndrome coronavirus 2 (SARS-CoV-2) [85,86].

## 6. Factors Influencing Vitamin D Status in Human Body

Vitamin D regulates calcium and phosphorus metabolism [87]. The compound plays an antagonistic position against the parathyroid hormone and stimulates PTH secretion by inducing calcium removal from bones. The level of vitamin D, and more precisely its active metabolite 1α, 25(OH)_2_D_3_, is under the control of PTH and depends on the content of calcium and phosphorus in the body [17]. Epidemiological studies suggest that even mild vitamin D deficiency increases PTH levels and may worsen bone condition. An increase in the value of calcidiol in serum from 20 to 32 ng/mL (50–80 nmol/L) improves calcium transport efficiency in the intestine by 45–65% [88]. In addition, calcitriol is regulated by fibroblast growth factor (FGF-23) produced by bone tissue. High blood calcitriol levels induce expression of the enzyme D-24-hydroxylase (CYP24A1), which may transform calcitriol into the inactive compound calcitroic acid [89]. This is a feedback system that regulates the concentration of active vitamin metabolic agents and prevents the body from hypervitaminosis D [22]. The relationship between PTH and low vitamin D level may be an important determinant mainly in the case of older people. With a decrease in vitamin D levels, calcium and phosphorus absorption becomes suboptimal. PTH level and alkaline phosphatase activity rise [90]. In people with low levels of 25(OH)D, and increase in PTH concentration, ionized calcium level may also be higher. This suggests that in similar populations vitamin D levels <25 ng/mL can be recognized as a cut-off point for starting vitamin D supplementation [91]. Alvarez et al. confirmed that 25(OH)D blood concentrations are independently associated with plasma glutathione, thiol redox status systems and inflammatory markers [92]. The metabolism of vitamin D also depends on magnesium, which acts as a cofactor for 25-hydroxylase, 1α-hydroxylase, and 24-hydroxylase. Moreover, the activity of the vitamin D binding protein (VDBP) is magnesium dependent [93]. Therefore, a low level of this mineral may change the link between 25(OH)D in serum and PTH [94]. Vitamin D anti-inflammatory properties were manifested by inhibiting the kappa B (NFκB) nuclear pathway. Serum 25(OH)D concentrations were positively associated with NFκB activity in peripheral blood mononuclear cells [95].

Vitamin D_3_ synthesis and its conversion into 25(OH)D_3_ depend on various socio-demographic and biochemical factors (Figure 2).

Among environmental factors, latitude, season, and residence significantly impact D_3_ skin synthesis. The status of vitamin D in the body also varies depending on ethnicity, gender, and age (Table 3).

The chances of vitamin D deficiency increase with age and lack of sun exposure. Vitamin D synthesis is most effective in the summer between 10 am and 3 pm in light-skinned people living below 40 degrees latitude [96]. In areas below the 34th parallel north, the amount of solar energy is large enough for vitamin D synthesis to occur throughout the whole year. Still, these regions’ inhabitants also have deficiencies, especially women living in the Middle East, who avoid the sun for religious reasons and cover the whole body, including the face [97,98].

There are several reports of the effect of residence on vitamin D levels. Griffin et al. [99], stated that different lifestyle led by rural residents, compared to urban residents, could be associated with a risk of vitamin D deficiency. The authors explained that rural dwellers are less exposed to sunshine because of changes in lifestyle as they now commute longer distances to work in the cities. Additionally, older rural dwellers may be less active due to geographic isolation and have less access to information on the benefits of good vitamin D health. It was shown that 25(OH)D levels in serum were significantly lower among rural residents in western Ireland, regardless of age. The deficiency of this vitamin was lower for women both in the city and in the countryside [99]. Moreover, Carpenter et al. [100] reported that concentrations of 25(OH)D depend on ethnicity, and are positively correlated with European ancestry (*p* < 0.001), and negatively correlated with African ancestry (*p* = 0.016).

Other natural factors that regulate vitamin D synthesis in the skin are skin pigmentation, sunscreen creams (SPF), and clothing. Large amounts of melanin in dark skin effectively block vitamin D synthesis. Melanin reduces the epidermis’ penetration by UVB, limiting the dermal synthesis of 25(OH)D [101]. Therefore, people of color require more prolonged exposure to the sun than Caucasians [102,103]. Sunscreen use with sun protection factor (SPF) 15 reduces cholecalciferol synthesis by up to 99% [104]. Most current health messages suggest several short sun exposure sessions in the summer to ensure sufficient vitamin D production. The guidelines indicate that 25% of the skin’s exposure for 15–30 min between 10 am and 3 pm, two or three times a week, may meet the demand for vitamin D in Caucasian people [96]. The evidence of Touvier et al. [105] also suggests that even low daily-life sun exposure behaviors contribute to increased vitamin D synthesis and improved 25(OH)D concentrations. Unfortunately, this does not apply to everyone. The amount that can be obtained in this way varies considerably from person to person and is influenced by several factors such as high BMI, race, ethnicity and genetic factors, namely polymorphisms in key genes of the vitamin D pathway, modulating vitamin D status [106]. However, during a study by Joh et al. [102], both sun exposure (≥20–30 min/day around noon) and 500 IU vitamin D_3_ supplementation significantly affected serum 25(OH)D levels compared to placebo. Supplementation alone, compared to UV exposure, caused a more significant increase in vitamin levels.

Deficiency of vitamin D is primarily a common health problem for the elderly. It is associated with decreased physical performance, and a higher risk of cognitive impairment, depression and anxiety. In people over 70, the amount of vitamin D precursor 7-dehydrocholesterol may be reduced by up to 60–75% [107]. Older people are vulnerable to many risk factors, such as reduced dietary intake, reduced exposure to sunlight (living in nursing homes), increased skin thickness, impaired intestinal absorption, and impaired hydroxylation process in liver and kidney. Older women had significantly lower levels of this vitamin [108]. However, Vitamin D deficiency occurs at all ages, mainly in children from 12 to 14 years old [109]. Numerous studies indicate a negative correlation of vitamin D concentration with parameters such as BMI, percentage of body fat, or WHR among children. It was proven that the rate of vitamin D deficiency in girls was higher than among boys. The level of this vitamin in the body could be influenced by adolescents’ behaviour and eating habits. It was shown that 25(OH)D levels correlated with obesity factors, mainly in boys. It is believed that girls have proportionally more subcutaneous fat, which poses a lower health risk than visceral and abdominal fat. Estrogens may also play a significant role in the metabolism of vitamin D in girls [109].

Obesity correlates with vitamin D deficiency through reduced bioavailability of this compound due to lower physical activity and lower sun exposure [110,111]. According to Monache et al., 25(OH)D concentration in serum correlated with BMI, regardless of season and age. Lower vitamin D values characterized women with obesity compared to women with healthy body weight and overweight in winter and summer [112,113]. Therefore, the developed guidelines recommend a twice-daily dose for obese people than the recommended dose for peers with appropriate body weight [114]. Low levels of vitamin D in the body may be caused by reduced bioavailability from the gastrointestinal tract due to impaired absorption or inability to use stored fat deposits in obese people. Another probable mechanism of decreased vitamin D concentration in this group is the increased synthesis of the active metabolite of vitamin D—1,25(OH)_2_D_3_ in the kidneys. This metabolite inhibits the production of 25(OH)D_3_ in the liver in a negative feedback mechanism [89].

Vitamin D deficiency correlates with diet. Cases of reduced vitamin D levels have been found in vegetarians, vegans, and macrobiotic nutritionists [115]. Interestingly, supplementation with high doses of vitamin D contributes to a change in the human intestinal microbiome. A decrease in the Bacteroidetes and Lactobacillus bacteria and an increase in Firmicutes and Bifidobacterium were observed [116].

Smoking may also be a predisposing factor for lowering vitamin D levels (<20 ng/mL). There is a hypothesis that smoking interferes with the conversion of 25(OH)D to 1,25(OH)_2_D, thus reducing calcium absorption in the gastrointestinal tract, leading to loss of bone mineral density, especially in men over 65 years of age [91,117].

Some medicines, e.g., antiepileptic drugs (phenytoin, phenobarbital), rifampicin, orlistat (used to treat obesity), cholestyramine, glucocorticosteroids, and immunosuppressants can reduce vitamin D levels [106].

DBP is a multifunctional protein in an ascetic fluid, plasma, the cerebrospinal fluid also found on the surface of numerous cells. It binds to the different forms of vitamin D including ergocalciferol, cholecalciferol, calcifediol, and active calcitriol [118]. Different physiological and pathological conditions can affect DBP levels, which in turn affect circulating concentrations of vitamin D forms. Carpenter et al. proved that 25(OH)D levels are positively correlated with circulating DBP (R = 0.25, *p* < 0.001) [100]. Moreover, a significant number of children (52.7%) with vitamin D deficiency had low DBP < 168 mg/L (*p* = 0.015) [119]. In pregnancy and liver diseases, the affinity of DBP for vitamin D metabolites may be decreased. It was reported that DBP level increases two-fold between the second and third trimesters while mean free 25(OH)D lowers and total 25(OH)D remains unchanged [120]. Liver diseases result in reductions in DBP. In patients with liver cirrhosis levels of free 25(OH)D were higher than in healthy subjects [121]. Therefore, alterations in DBP levels should be considered as potential confounders of the interpretation of plasma total 25(OH)D concentrations. Bikle and Schwartz [122] suggest that assessment of vitamin D status might be improved by measuring free 25(OH)D instead of, or in addition to, total 25(OH)D.

Gene mutations are important factors affecting the synthesis and metabolism of vitamin D. They include polymorphisms of CYP450 isoenzymes, protein transporters, and VDR. In the case of impaired catabolism of 25(OH)D and 1,25(OH)_2_D caused by mutations in the CYP24A1 gene encoding 24-hydroxylase, or the fact of excessive synthesis of 1,25(OH)_2_D resulting indirectly from the mutation in the SLC34A1 gene encoding the sodium co-transporter phosphate (NaPi-IIA) in the kidney, elevated calcitriol concentration is observed [123]. The risk of hypervitaminosis D in both cases is increased with the use of prophylactic doses of vitamin D. The polymorphism of the gene responsible for the metabolic pathway of vitamin D (GC gene-rs2282679) encodes DBP, which is responsible for regulating the transport of 25(OH)D and 1,25(OH)_2_D to tissues. Studies in various populations have shown a strong association between GC-sr2282679 and vitamin D status at lower 25(OH)D levels in people with the CC genotype, suggesting the effect of vitamin D on length telomeres. The fact that people with CC genotype had lower LTL (leukocyte telomere length) than people with AA genotype confirms the hypothesis regarding the long-term effect of vitamin D on telomere shortening [124]. However, in many bigger population studies, no relationship was found between 25(OH)D serum concentrations and LTL [125]. The VDR gene is located on the long arm of chromosome 12 (12q12-14) and has about 200 single nucleotide polymorphisms (SNPs). The Bsm variant of the VDR gene was associated with hypertriglyceridemia and may be predisposed to metabolic syndrome (MetS). The VDR TaqI TT and BsmI BB + Bb genotypes were associated with lower 25(OH)D levels in the metabolic syndrome group [126]. Moreover, in obese patients with vitamin D deficiency, carriers of polymorphic alleles showed significant lower levels of serum 25(OH)D and higher HOMA-IR (the homeostasis model assessment of insulin resistance; *p* = 0,04), blood pressure levels (*p* < 0.001) and lipid parameters compared to those with the wild type homozygotes (*p* = 0.02) [127]. Cdx2 polymorphism may be a potential biomarker in vitamin D treatment for breast cancer, irrespective of VDR receptor expression. The B allele or Bb VDR genotype presence may be a risk factor for breast cancer development [128].

Several studies proved that after supplementation with 25(OH)D_3_, vitamin D serum levels appeared to be genetically modified [118,129,130,131,132]. Carriers of the common AA genotype of rs4588 polymorphism better respond to vitamin D supplementation [118]. People carrying allele G of CYP2R1 had a higher vitamin D concentration after nine weeks of supplementation [131]. Moreover, CYP2R1 and CYP24A1 DNA methylation levels could potentially be biomarkers of variability in vitamin D responses. People with high levels of DNA methylation in two genes may need high vitamin D supplementation to achieve optimal 25(OH)D serum levels [132].

## 7. Conclusions

With the discovery of vitamin D-binding receptors in various cells and tissues, it was proved that vitamin D affects the proper functioning of many systems in the body including the immune system, the cardiovascular system, vision, the growth and division of cells and the formation of blood vessels. Numerous studies have confirmed the relationship of low concentrations of vitamin D and the occurrence of many diseases including rheumatoid arthritis, metabolic disorders, cancer and depression. Clinical interest in the physiological importance of vitamin D forms and their possible roles in pathophysiological processes requires an appropriate method for their measurement. It is increasingly apparent that laboratories using chromatographic based methods for 25(OH)D analysis, should separate the epimeric form of 25(OH)D_3_ in order to provide completer and more accurate 25(OH)D results. Therefore, identification of a rapid, simple, reliable, and cost-effective method for the determination of vitamin D status could help in monitoring disease progression and therapeutic strategies, improving quality of care. When analyzing vitamin D status, various socio-demographic, biochemical and genetic factors influencing vitamin D synthesis and metabolism should be taken into account.

## Figures and Tables

**Figure 1 nutrients-12-02788-f001:**
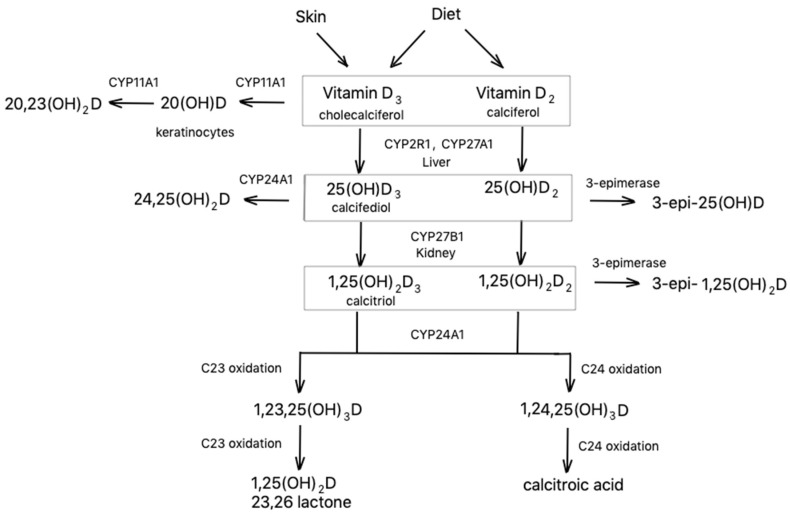
Metabolism of vitamin D.

**Figure 2 nutrients-12-02788-f002:**
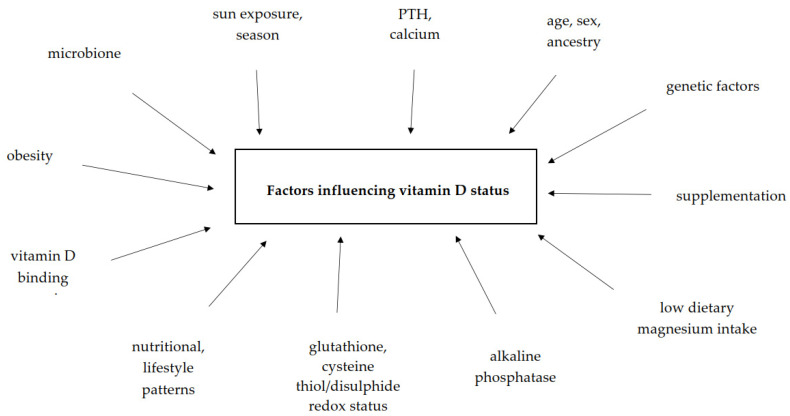
Determinants of vitamin D status in the human body.

**Table 1 nutrients-12-02788-t001:** Recent liquid chromatography coupled with tandem mass spectrometry (LC-MS/MS) methods for analysis of forms and metabolites of vitamin D.

Analyzed Compounds	Matrix	Method	Sample Preparation	Chromatographic Conditions	LOQ (ng/mL)	Ref.
25(OH)D_3_, 25(OH)D_2_	Plasma	LC-MS/MS	alkaline hydrolysis of lipid esters followed by SPE	Column: Supelcosil LC-18-S (250 × 4.6 mm, 5 µm) Mobile phase: 0.1% formic acid in methanol, 0.1% formic acid in methanol/water (50:50, *v/v*), gradient elution.	25(OH)D_3_: 3.2 25-OH-D_2_: 3.4	[39]
25(OH)D_2_, 25(OH)D_3_	Dried blood spots serum	LC-MS/MS	protein precipitation with methanol, LLE with hexane	Column: Varian Pursuit 3u PFP (50 × 2.0 mm, 3 µm). Mobile phase: 0.1% formic acid and 2 mM ammonium acetate, isocratic elution.	n.a.	[40]
25(OH)D_2_, 25(OH)D_3_	Serum	LC-MS/MS	Protein precipitation with methanol and ZnSO_4_	Column: PFP (150 × 2.1 mm, 2.5 µm). Mobile phase: 0.1% formic acid in water and methanol, gradient elution.	25(OH)D_3_: 0.9 25(OH)D_2_: 1.03	[41]
3α-25(OH)D_3_, 3β-25(OH)D_3_	Dried blood spots serum	LC-MS/MS	1. Dried blood spots: extraction 2. serum: SLE	Kinetex PFP F5 100A column (100 × 2.1 mm, 2.6 µm). Mobile phases: (A) water 0.1% formic acid and (B) methanol 0.1% formic acid, gradient elution.	3α-25(OH)D_3_: 0.1 3β-25(OH)D_3_: 1.0	[42]
25(OH)D_3_, 3-epi-25(OH)D_3_	Serum	LC-MS/MS	protein precipitation with acetonitrile	Column: Luna C18 (250 × 4.6 mm, 2.3 µm) Mobile phase: 0.1% formic acid in water and in acetonitrile, isocratic flow	n.a.	[43]
25(OH)D_3_, 3-epi-25(OH)D_3_	Plasma (cord blood)	LC-MS/MS	LLE with hexane	Column: PFP (150 × 2 mm, 3 µm).	25(OH)D_3_: 1.4 3-epi-25(OH)D_3_: 1.4	[44]
25(OH)D_2_ 25(OH)D_3_, 24,25(OH)_2_D_3_, 3-epi-25(OH)D_3_	Serum	LC-MS/MS	LLE with hexane	Column: Kinetex F5 (50 × 2.1 mm, 1.7 µm). Mobile phase: water and acetonitrile with 0.1% formic acid, isocratic elution.	n.a.	[45]
24,25(OH)_2_D_3_ 25(OH)D_3_ 3-epi-25(OH)D_3_ 25(OH)D_2_	Serum	LC-MS/MS	0.2 mM aqueous zinc sulfate, methanol. LLE with ethyl acetate and hexane	Column: F5 (100 mm × 2.1 mm, 2.7 µm). Mobile phase: 0.1% formic acid, 2 mM ammonium acetate in water, and 0.1% formic acid, 2 mM ammonium acetate in methanol, gradient elution	n.a.	[46]
D_2_-S, D_3_-S 25(OH)D_2_-S 25(OH)D_3_-S	Serum, breast milk	LC-MS/MS	Sample precipitation with acetonitrile	Column: EC-C18 (15 × 2.1mm, 2.7 µm), Mobile phase: water with 10mM ammonium formate and methanol with 10mM ammonium formate, gradient elution.	Milk/serum D_2_-S: 0.002/0.003 D_3_-S: 0.003/0.003 25(OH)D_2_-S: 0.003/0.004 25(OH)D_3_-S: 0.003/0.004	[47]
25(OH)D_3_-S, 25(OH)D_3_-G	Serum, plasma	LC-MS/MS	SPE, derivatization with DAPTAD	Column: Hypersil Gold (2.1 × 100 mm, 1.9 μm) Mobile phase: 5 mM ammonium acetate, acetonitrile, gradient elution.	25(OH)D_3_-S: 2.5 25(OH)D_3_-G: 1.73	[48]
25(OH)D_3_, 25(OH)D_2_, 3-epi-25(OH)D_3_, 1α,25(OH)_2_D_3_, 23,25(OH)_2_D_3_, 24,25(OH)_2_D_3_, 3-epi-25(OH)D_2_, 1α,25(OH)_2_D_2_, 1α,24(OH)_2_D_2_, D_2_, D_3_	Serum	LC-MS/MS	SLE	Column: Lux Cellulose- 3 chiral column (100 × 2 mm, 3 µm). Mobile phase: methanol/water/0.1% formic acid, gradient elution	n.a.	[16]
D_3_, 25(OH)D_3_, 24R,25(OH)2 D_3_, 1α,25(OH)2 D_3_, 4β,25(OH)2 D_3_	Plasma	LC-MS/MS	LLE with ethyl acetate	Column: Hypersil Gold (2.1 × 100 mm, 1.9 µm). Mobile phase: acetonitrile/water (0.1% formic acid)	1a,25(OH)_2_D_3_: 0.025 24R,25(OH)_2_D_3_: 0.05 25OHD_3_: 0.05	[49]
25(OH)D_2_, 25(OH)D_3_, 3-epi-25(OH)D_3_, 24,25(OH)2D_3_	Serum	LC-MS/MS	protein precipitation with acetonitrile and ZnSO4	Column: Kinetex PFP 100 Å (100 × 2.1 mm, 2.6 μm). Mobile phase: water and methanol, both with 0.1% formic acid, gradient elution.	24,25(OH)_2_D_3_: 0.5 25(OH)D_3_: 1.1 epi-25(OH)D_3_: 1.1 25(OH)D_2_: 1.7	[50]
24,25(OH)_2_D_3_, 24,25(OH)_2_D_2_, 25(OH)D_3_, 25(OH)D_2_	Serum	LC-MS/MS	SLE	Column: core-shell C18 (50 × 2.1 mm, 2.6 μm). Mobile phase: water and methanol, both with 0.2 mM methylamine in 0.1% formic acid, gradient elution.	24,25(OH)_2_D_3_, 25(OH)D_3_, 25(OH)D_2_: 0.04 24,25(OH)_2_D_2_: 0.34	[51]
24,25(OH)_2_D_3_, 24,25(OH)_2_D_2_, 1,25(OH)_2_D_2_, 25(OH)D_3_, 3-epi-25(OH)D_3_, 25(OH)D_2_, D_2_, D_3_	Serum	2D-LC-MS-MS	SPE	Columns: Poroshel 120 EC-C18 (50 × 4.6 mm, 2.7 µm), Pursuit PFP (100 × 4,6 mm, 3 µm). Mobile phase: water and methanol, both with 5 mM ammonium formate, 20:80, gradient elution.	24,25(OH)_2_D_3_, 24,25(OH)_2_D_2_: 0.03 1,25(OH)_2_D_2_: 0.05 25(OH)D_3_, 25(OH)D_2_: 0.25 3-epi-25(OH)D_3_: 0.3 D_2_, D_3_: 0.1	[52]
1,25(OH)_2_D_2_, 1,25(OH)_2_D_3_	Serum	2D ID-UPLC-MS/MS	Immuno-extraction	Column 1: C_4_ BEH300 (50 × 2.1 mm, 1.7 µm) Mobile phase 1: water/water with 0.2%. formic acid/acetonitrile, 75:20:5 (*v/v/v*), gradient elution. Column 2: C_18_ BEH (100 × 2.1 mm, 1.7 µm) Mobile phase 2: 60:40 (*v/v*) water with 0.2% formic acid/acetonitrile, gradient elution.	1,25(OH)_2_D_2_: 0.0015 1,25(OH)_2_D_3_: 0.0014	[19]
D_3_, D_2_, 25(OH)D_3_, 1(OH)D_3_, 1(OH)D_2_, 24,25(OH)_2_D_3_, 1,25(OH)_2_D_2_, 1,25(OH)_2_D_3_	Plasma	UPSFC-MS	LLE with acetonitrile	Columns: Torus 2-picolylamine, Torus diethylamine, Torus high density diol, Torus 1-aminoanthracene, fluorophenyl (each column: 100 × 3 mm, 1.7 µm), HSS C18SB (100mm × 3mm, 1.8 µm) Mobile Phase: CO_2_, methanol, gradient elution.	D_3_: 5.43, D_2_: 7.25 25(OH)D_2_: 17.22 25(OH)D_3_: 6.56 1(OH)D_3_: 7.75 1(OH)D_2_: 18.11 24,25(OH)_2_D_3_: 1.19 1,25(OH)_2_D_2_: 6.18 1,25(OH)_2_D_3_: 7.57	[13]
25(OH)D_3_, 25(OH)D_2_, 24(OH)D_2_, D_2_, D_3_, 3-epi-25(OH)D_3_, 1α,25(OH)_2_D_3_, 24R,25(OH)_2_D_3_, 23R,25(OH)_2_D_3_, 1α,25(OH)_2_D_3_-PTAD, 24,25(OH)_2_D_3_-PTAD	Serum	UPSFC- MS/MS	SLE	Column: Lux cellulose-3 chiral column (150 × 3 mm, 3 μm) and UPC2 BEH column (100 × 3 mm, 1.7 μm). Mobile phase: CO_2_ and 0.1% formic acid in methanol with a make-up solvent of 0.1% formic acid, gradient elution	1α,25(OH)_2_D_3_: 0.08	[53]

n.a.—not available; LLE—liquid-liquid extraction; SLE—supported liquid extraction; SPE—solid phase extraction; PTAD—4-phenyl-1,2,4-triazoline- 3,5-dione; SFC—supercritical fluid chromatography; 25(OH)D_3_-S—sulphate of 25(OH)D_3_; 25(OH)D_3_-G—glucuronide of 25(OH)D_3_; UPLC—ultra-performance liquid chromatography; UPSFC—ultra-performance supercritical fluid chromatography; LOQ—limit of quantitation.

**Table 2 nutrients-12-02788-t002:** Analysis of vitamin D status in various diseases.

Analyzed Compounds	Matrix	Disease	Studied Group (N)	Conclusions	Ref.
25(OH)D_3_ 3-Epi-25(OH)D_3_, 25(OH)D_2_, 24,25(OH)_2_D_3_, 1,25(OH)_2_D_3_	Synovial fluid serum	RA	20 patients with rheumatoid arthritis (RA), 7 patients with resolving reactive arthritis (ReA), 23 healthy controls	significantly lower serum 3-epi-25(OH)D_3_ in RA (median 0.788 ng/mL, *p* < 0.01) and ReA patients (median 0.361 ng/mL, *p* < 0.05) relative to healthy controls	[57]
vitamin D	serum	RA	149 patients with RA	statistically significant improved mean disease activity index in RA patients after supplementation (*p* = 0.002) improved serum vitamin D levels (from 10.05 ± 5.18 to 57.21 ± 24.77 ng/mL, *p* < 0.001) during the treatment.	[59]
1,25(OH)_2_D 25(OH)D	serum	psoriasis	122 patients with psoriasis	Inverse relationship found between 1,25(OH)2D and: visceral adipose (β = −0.43, *p* = 0.026). aortic vascular uptake of 18F-fluorodeoxyglucose (β = −0.19, *p* = 0.01) non-calcified coronary plaque burden (β = −0.18, *p* = 0.03)	[61]
25(OH)D_3_	serum	metabolic disorders	92 subjects deficient in vitamin D; 48 with vitamin D supplementation and 44 without supplementation	Higher serum level of vitamin D after three-month supplementation with 2000 IU vitamin D (*p* < 0.001) Association of higher exposure to vitamin D and decreased level of oxidative DNA damage in lymphocytes (*p* < 0.05), increased HDL, decreased HOMA-IR, TG/HDL ratio.	[64]
25(OH)D	serum	breast and prostate cancer	15.748 breast cancer cases, 22.898 prostate cancer cases	No association between 25(OH)D and risk of breast cancer, estrogen receptor, prostate cancer or the advanced cancer subtype.	[69]
25(OH)D	blood *	colorectal cancer	5706 colorectal cancer participants, 7107 controls	Association of deficient 25(OH)D (<30 nmol/L) with 31% higher colorectal cancer risk (RR = 1.31). Association of 25(OH)D concentrations of 75–<87.5 and 87.5–<100 nmol/L with 19% (RR = 0.81) and 27% (RR = 0.73) lower risk of colorectal cancer.	[70]
Vitamin D	serum	Cancer cardiovascular disease	25,871 (12,927 after supplementation, 12,944 after placebo)	After 1-year supplementation of 2 000 IU vitamin D: 40% increase in mean 25(OH)D levels no result in a lower incidence of invasive cancer or cardiovascular events.	[71]
1,25(OH)_2_D_3_	serum	Diabetic Retinopathy	66 diabetic patients, 20 nondiabetic healthy patients	Lower mean serum 1,25(OH)2D_3_ and 25(OH)D concentrations in diabetic patients (*p* < 0.001) lower 1,25(OH)2D_3_ concentrations in patients with diabetic retinopathy negative correlations between 1,25(OH)2D_3_ and age (*p* < 0.01) and duration of diabetes (*p <* 0.05).	[72]
25(OH)D	serum	chronic obstructive pulmonary disease	278	Association of vitamin D deficiency (<50 nmol/L) with: reduced % predicted forced expiratory volume in one second (p for trend = 0.06) reduced % predicted forced vital capacity (*p* for trend = 0.003).	[73]
25(OH)D	serum	metabolic syndrome	559 Chinese subjects at elevated risk of metabolic syndrome	Lower 25(OH)D levels in participants with obesity, high triglycerides, type 2 diabetes, or MS (all *p* < 0.05) 2.5 times higher incidence of MS in participants in the lowest 25(OH)D tertile compared to those in the highest tertile (OR 2.48; *p* < 0.05).	[74]
25(OH)D	Dried blood spots	multiple sclerosis	521 patients with multiple sclerosis, 972 controls	Highest multiple sclerosis risk among individuals with 25(OH)D concentration < 20.7 nmol/L and lowest among those with 25(OH)D levels ≥48.9 nmol/L (OR 0.53).	[75]
25(OH)D	serum	Bronchiolitis	50 infants with bronchiolitis, 31 controls	Significantly lower the mean serum 25(OH) vitamin D in patients with bronchiolitis (*p* = 0.003). Non-significant negative correlation of serum IgE with serum 25(OH) vitamin D (r = −0.141, *p* ≥ 0.05).	[76]
25[OH]D_3_	serum	food allergy	5276, 269 nonallergic at age 1 y, 338 food allergic at age 1 y, 50 egg tolerant at age 2 y, 55 egg allergic at age 2 y	Association of low serum 25(OH)D_3_ level (≤50 nM/L) at age 1 years with: food allergy (OR 12.6) among infants with the GG genotype for DBP (OR 6.0) but not in those with GT/TT genotypes (OR, 0.7, *p* = 0.014).	[77]
25(OH)D_3_	serum	chronic kidney disease, depression	533 Koreans participants	Higher prevalence of depression in chronic kidney disease patients with vitamin D deficiency (32.5% vs. 50.0% without deficiency, *p* < 0.001). Vitamin D deficiency as a significant independent predictor of depression (OR 6.15; *p* = 0.001).	[78]
Vitamin D	serum	Depression	5006	More depressive symptoms in men and women with deficient (<30 nmol/L) vs. adequate (≥50 nmol/L) vitamin D status Higher incidence of current major depressive disorder (OR 2.51) in men with deficient vitamin D status.	[79]
25(OH)D	serum	Post-Traumatic Stress Disorder (PTSD)	1653	25(OH)D levels inversely associated with PTSD (OR: 0.96; *p* = 0.044) Vitamin D deficiency positively associated with PTSD (OR = 2.02; *p* = 0.028).	[80]

* not specified if plasma or serum was used for 25(OH)D measurements. HOMA-IR—homeostasis model assessment of insulin resistance.

**Table 3 nutrients-12-02788-t003:** Factors influencing vitamin D status in human body.

Factors	Studied Group (N)	Conclusions	Ref.
Parathyroid hormone (PTH), calcium	2259 adults (18–68 years old).	Significant correlations between Ca^2+^ and PTH (*r* = −0.223, *p* < 0.001), 25(OH)D and PTH (*r* = −0.178, *p* < 0.001) and between PTH and age (*r* = 0.322, *p* < 0.001) were found.	[17]
calcium, PTH, alkaline phosphatase	58 children and adolescents	A positive and significant correlation was found between dietary calcium and vitamin D (*r* = 0.77, *p* < 0.001).	[90]
sun exposure (<30 min and ≥30 min per week) PTH ionized calcium	1339 ≥18 years old	The median of 25(OH)D <10 ng/mL associated with hypercalcemia. The levels of 25(OH)D were higher in women who received >30 min of sun exposure per week, and who claimed to use sunscreen <3 times/week (*p* ≤ 0.001).	[91]
glutathione and cysteine thiol/di-sulfide redox status	693 adults (449 females, 244 males)	Serum 25(OH)D was positively associated with plasma GSH and negatively associated with plasma redox potentials—Eh GSSG and Cys (*p* < 0.001 for all).	[92]
low dietary magnesium intake	57 (22–65 years old, BMI 25–45 kg/m^2^)	Higher serum levels of 25(OH)D were negatively associated with lower PTH in the high magnesium intake group (*p* = 0.041). A positive relationship between 25(OH)D and serum adiponectin concentrations was observed in the high magnesium intake group (*r* = 0.532, *r* = 0.022). Serum interleukin-6 concentrations were negatively associated with 25(OH)D levels (*r* = −0.316, *p* = 0.017).	[94]
nuclear factor *kappa*-B activity	49	In healthy adults, 25(OH)D concentrations were positively associated with NFκB activity in peripheral blood mononuclear cells (r = 0.48, *p* = 0.0008).	[95]
place of residence: urban area, rural area, season, sex	17,590 (urban *n* = 4824; rural *n* = 12,766)	Serum 25(OH)D concentrations were lower among rural compared to urban dwellers and depend on sex (*p* < 0.001) and age (for urban *p* < 0.001, for rural *p* < 0.001).	[99]
Ancestry, vitamin D binding protein	750 healthy children (6–36 months old)	25(OH)D levels are positively correlated with circulating DBP (R = 0.25, *p* < 0.001). Circulating 25(OH)D was positively correlated with European ancestry (*p* < 0.001), and negatively correlated with African ancestry (R = −0.09, *p* = 0.016).	[100]
vitamin D supplementation, sun exposure (>20 min/day during summer, and >30 min/day during fall)	50: sun 50: oral vitamin D_3_ 50: placebo	Increases in serum 25(OH)D were greater with oral vitamin D_3_ than with sun exposure (difference in changes = 6.3 ng/mL, 95% CI: 4.3, 8.3). 54.2% participants in the oral vitamin D_3_, 12.2% in the sun exposure and 4.3% controls achieved serum 25(OH)D concentrations ≥20 ng/mL	[102]
sun exposure (0–1 h/day, 1–3 h/day, and >3 h/day), dietary intake	1084 adults	The odds of having 25(OH)D <20 ng/mL significantly decreased with being very active (OR 0.55), increasing length of sun exposure (1–3 h/day (OR 0.59), >3 h/day (OR 0.36)), and skin color (light to medium skin (OR 0.47), fairly dark skin color (OR 0.34) and dark or very dark skin color (OR 0.34)), compared to respective baseline levels.	[103]
sex	50 > 65 years old	Significant association between low vitamin D level and female gender (*p* = 0.024), advanced age (*p* = 0.026), no-sun exposure jobs (*p* = 0.001) and nursing home residency.	[108]
adiposity, age sex	10,696 at 6–18 years old	The prevalence rates of vitamin D deficiency and insufficiency were higher in girls (31% and 83.4%, respectively) than in boys (22.8% and 78.7%, respectively). Fat mass index and fat mass percentage were inversely associated with 25(OH)D concentrations, particularly in boys (*p* < 0.05). The association of age with vitamin D had L-shape with a threshold age of 14.	[109]
adiposity	163 obese	Serum 25(OH)D concentrations were negatively associated with percent body fat (%BF) (*p* = 0.003), positively associated with skeletal muscle mass (SMM) (*p* = 0.03).	[110]
adiposity	797	Mean 25(OH)D levels were significantly higher in normal weight and overweight males compared to obese males (*p* < 0.05) and in overweight females compared to obese females (*p* < 0.05). BMI, waist circumference, and waist-to-height ratio were inversely correlated with 25(OH)D levels (*p* < 0.001).	[111]
BMI, adiposity	women with age 19–80	25(OH)D concentration was dependent on season. BMI demonstrated the highest significant inverse correlation with serum 25(OH)D values (*p* < 0.001), independently from season and age.	[112]
age, sex, obesity, season, latitudes, lifetime sun exposure (scores: 1–10), physical activity, ancestry	1828 Caucasian middle-aged men and women	Vitamin D status was lower among women (*p* < 0.0001), older subjects (*p* = 0.04), obese or underweight subjects (*p* < 0.0001), subjects who lived at higher latitudes (*p* < 0.0001), and those whose blood draw occurred in early spring (*p* < 0.0001). Vitamin D status was higher among subjects who were more physically active (*p* < 0.0001), who had higher scores of usual sun exposure (*p* < 0.0001), those with higher Fitzpatrick photo-type (*p* = 0.03).	[105]
nutritional and lifestyle patterns	116	Vitamin D deficiency was associated with higher systolic ambulatory and daytime blood pressure monitoring (*p* = 0.01 and *p* = 0.02, respectively), lower step counts, lower urinary calcium, and higher fat mass. Milk intake (*p* = 0.009) and fish (*p* < 0.001) were lower in the deficient.	[113]
vitamin D supplementation, microbiome	50 adolescent girls before and after vitamin D supplementation	The mean (±SD) of serum vitamin D level at baseline was 11 ± 9 ng/mL and after high dose vitamin D supplementation it increases to 40 ± 17 ng/mL (*p* < 0.001). A high dose supplementation of vitamin D alter the human gut microbiome composition: Bacteroidetes and Lactobacillus fell by 72% and 24% respectively, whilst Firmicutes and Bifidobacterium were increased by1.5 and 1.2 fold after supplementation.	[116]
Vitamin D binding protein	210 children (1–5 years old)	25(OH)D levels correlated positively with DBP (r = 0.298, *p* = 0.0001). 52.7% of children with vitamin D deficiency had low DBP (*p* = 0.015). Despite adequate sun exposure, 43% of children showed vitamin D deficiency and 56.6% had low DBP levels.	[119]
Vitamin D binding protein	368 pregnant women	Free 25(OH)D lowers by 12% in the 3rd trimester comparing to the 1st trimester (*p* < 0.05) whereas total 25(OH)D was not decreased. DBP rises with gestational age.	[120]
Vitamin D binding protein, race	1661 adults (healthy, prediabetic, pregnant, cirrhotic, nursing home residents)	Levels of free 25(OH)D were higher in patients with cirrhosis (*p* < 0.0033) while DBP concentrations were lower than in other groups and differed between whites and blacks (*p* < 0.0033) and between DBP haplotypes (*p* < 0.0001).	[121]
rs12785878, rs10741657, rs6013897, rs2282679	461 (33–79 years old)	Participants with CC genotype (rs2282679) had shorter age- and sex-adjusted mean leukocyte telomere length (LTL) than those with AC and AA genotypes (*p* < 0.05). Serum 25(OH)D concentrations were not associated with LTL.	[124]
FokI BsmI TaqI Cdx2	237 participants with metabolic syndrome (MetS), 376 controls	VDR TaqI TT, and BsmI BB + Bb genotypes were associated with lower 25(OH)D levels (*p* < 0.05) in comparison to TaqI Tt, and BsmI bb genotypes in the MetS group. Cdx2 GG genotype was associated with higher waist circumference compared with the AG genotype in all subjects (*p* < 0.05). BB + Bb genotypes of the VDR BsmI had significantly increased the odds ratio of hypertriglyceridemia when compared with the bb genotype (OR 1.87, *p* = 0.022).	[126]
BsmI Apa-I TaqI obesity	201 obese women with vitamin D deficiency; 249 controls	In obese with vitamin D deficiency, carriers of polymorphic alleles showed significant lower levels of serum 25(OH)D and higher HOMA-IR (the homeostasis model assessment of insulin resistance; *p* = 0.04), blood pressure levels (*p* < 0.001) and lipid parameters compared to those with the wild type homozygotes (*p* = 0.02).	[127]
receptor gene BsmI (A/G) polymorphism	60 females with breast cancer (BC)	25(OH) vitamin D levels were significantly lower in the patients with BC compared to controls (*p* ≤ 0.001). Carriers of Bb genotype had 4.6 times increased risk of developing breast cancer when compared to other genotypes.	[128]
BsmI polymorphism of the VDR gene, supplementation	40 elderly women with vitamin D insufficiency	Supplementation with a vitamin D_3_ megadose reduced inflammatory markers and increased the total antioxidant capacity in elderly women with vitamin D insufficiency (*p* = 0.03). The 25(OH)D (*p* = 0.0001), PTH (*p* = 0.009), us-CRP (*p* = 0.007) and α1-acid glycoprotein (*p* = 0.005) levels of elderly patients with the BB/Bb genotype were more responsive to supplementation compared with those with the bb genotype.	[129]
CYP2R1	27 children with rickets, 50 unrelated subjects	After supplementation with 50,000 IU of vitamin D_2_ or vitamin D_3_, heterozygous subjects for the L99P and K242N mutations had lower increases in serum 25(OH)D than control subjects.	[130]
rs4588, supplementation	619 healthy adolescent girls	Polymorphism of rs4588 was associated with serum 25(OH)D both at baseline (*p* = 0.03) and after supplementation (*p* = 0.008). The subjects with common AA genotype were a better responder to vitamin D supplementation than GG.	[118]
CYP2R1 (rs10766197), supplementation	253 healthy girls	Subjects who had homozygous major allele GG showed two-fold higher response in serum 25(OH)D than carriers of the uncommon allele A (OR = 2.1, *p* = 0.03).	[131]
DNA methylation levels of CYP2R1, CYP24A1, CYP27A1, CYP27B1	446 women supplemented with calcium and vitamin D	For CYP2R1, baseline DNA methylation levels at eight CpG sites were negatively associated with the 12-month increase in serum 25(OH)D (*p* < 0.05). For CYP24A1, baseline DNA methylation levels at −342C and −293C were negatively associated with vitamin D response variation (*p* = 0.011, *p* = 0.025, respectively).	[132]

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
