# Peer review of "Clinical Significance of Analysis of Vitamin D Status in Various Diseases"

_nutrients, 2020, doi:10.3390/nu12092788_

Round 1

Reviewer 1 Report

The review article by Kowalowka et al is a well written article on the vitamin D metabolism, metabolites and clinical implications. Some suggestions are included below to improve clarity

  1. Section 2: Please include a figure for vitamin D metabolism. Also, define all metabolites, including all normal ranges. Discuss the importance of different vitamin D metabolites
  2. Section 4: Please include subsections for different methodologies for assessing 25OHD. Define and expand RIA, EIA, CLIA, and CPBA and LCMS techniques
  3. Table 2: Clarify blood vs serum. Use bullets for Conclusions. Some of them are bulleted, some seem to be lengthy with full sentences.
  4. Section 6: A figure may be good in this section.
  5. Table 3: Please add influence of adiposity and age
  6. The influence of Vitamin D binding protein should be discussed and also included in the tables

Author Response

1. Section 2: Please include a figure for vitamin D metabolism. Also, define all metabolites, including all normal ranges. Discuss the importance of different vitamin D metabolites

Answer: The figure for vitamin D metabolism has been added to the manuscript. In Section 2 we have defined all metabolites, discussed their importance and provided available ranges for serum concentrations. 

2. Section 4: Please include subsections for different methodologies for assessing 25OHD. Define and expand RIA, EIA, CLIA, and CPBA and LCMS techniques

Answer: Section 4 has been divided into two subsections covering immunoassay techniques and HPLC-MS/MS methods. The abbreviations were defined and expanded.

3. Table 2: Clarify blood vs serum. Use bullets for Conclusions. Some of them are bulleted, some seem to be lengthy with full sentences.

Answer: The above remarks were applied to Table 2.

4. Section 6: A figure may be good in this section.

Answer: The figure has been added.

5. Table 3: Please add influence of adiposity and age

Answer: Influence of adiposity and age has been added to Table 3 and discussed in the manuscript.

6. The influence of Vitamin D binding protein should be discussed and also included in the tables

Answer: Influence of vitamin D binding protein has been added to Table 3 and discussed in the manuscript.

Reviewer 2 Report

Review of manuscript # nutrients-899502

Clinical significance of analysis of vitamin D status in various diseases

The manuscript by Kowalówka et al. provides a review of the literature concerning the assessment of vitamin D status during various chronic conditions. The techniques utilized and confounding factors are particularly highlighted. Overall, it is a well-written review that covers a lot of research. Issues exist concerning the description of results and in-text citations.

Minor comments

Line 67 – 90% should be spelled out

Lines 88 – 89 – The formation of lumisterol and tachysterol should be mentioned

Line 114 – Grammar must be corrected

Lines 114 – 116 – Later in lines 176-178 and 226-227, there Is evidence it may be useful for RA

Line 140 – What % of C-3 epimers is commonly present versus total non-epimer 25(OH)D?

Line 150 - Grammar must be corrected

Line 177 – Should be 1,25(OH)2D3

Line 201 – Needs references

Line 220-223 - Grammar must be corrected

Lines 222 – 223 – Did this supplementation impact incidence?

Lines 262-264 – Where is the reference? If there are no peer-reviewed publications, this sentence should be removed

Line 267 – Better to say it “regulates calcium and phosphorus metabolism.”

Lines 285-286 – For which enzymes related to vitamin D metabolism is it a cofactor?

Line 301 – Please explain what you mean by “difficult lifestyle”

Line 319 – How much sun exposure?

Table 3 – If the study involved sun exposure treatment, briefly state details concerning the sun exposure such as length of time.

Line 403 – “complete and accurate”

Major comments

  • The Abstract should include mention of some of the general findings (i.e. Results) found in the review
  • The authors must be more conservative in their description of the results. For example, on lines 200-201, it is stated that “insulin resistance syndrome … and vitamin D deficiency contribute greatly…cancer.” It would be best to state the “evidence suggests” or it “may contribute”. This is just one example of numerous instances within the manuscript. In the revisions, please specify the line #’s where these corrections were made
  • In-text citations should be included at first mention of a fact. Throughout the manuscript, many facts are stated without a reference.

Author Response

Line 67 – 90% should be spelled out

Answer: It has been corrected.

Lines 88 – 89 – The formation of lumisterol and tachysterol should be mentioned

Answer: It has been added.

Line 114 – Grammar must be corrected

Answer: It has been corrected.

Lines 114 – 116 – Later in lines 176-178 and 226-227, there Is evidence it may be useful for RA

Answer: The sentence has been corrected.

Line 140 – What % of C-3 epimers is commonly present versus total non-epimer 25(OH)D?

Answer: It has been added: "may constitute even 50% of the  25(OH)D content in adults"

Line 150 - Grammar must be corrected

Answer: It has been corrected

Line 177 – Should be 1,25(OH)2D3

Answer: It has been corrected.

Line 201 – Needs references

Answer: References have been provided.

Line 220-223 - Grammar must be corrected

Answer: It has been corrected.

Lines 222 – 223 – Did this supplementation impact incidence?

Answer: The sentence has been corrected to be clearer:

"In the study of Manson et al., after one year of supplementation with 2000 IU of vitamin D, the average levels of 25-hydroxyvitamin D increased by 40%. However, the vitamin D supplementation did not result in a lower incidence of invasive cancer [71]."

Lines 262-264 – Where is the reference? If there are no peer-reviewed publications, this sentence should be removed

Answer: References have been provided.

Line 267 – Better to say it “regulates calcium and phosphorus metabolism.”

Answer: It has been corrected.

Lines 285-286 – For which enzymes related to vitamin D metabolism is it a cofactor?

Answer: The information on enzymes has been added: " a cofactor for 25-hydroxylase, 1α-hydroxylase, and 24-hydroxylase. Moreover, activity of vitamin D binding protein (VDBP) is magnesium dependent [93]"

Line 301 – Please explain what you mean by “difficult lifestyle”

Answer: It has been explained in the text: "The authors explained that rural dwellers are less exposed to the sunshine because of change in lifestyle as they are now commuting longer distances to work in the cities. Additionally, older rural dwellers may be less active due to geographic isolation and have less access to the information on the benefits of good vitamin D health"

Line 319 – How much sun exposure?

Answer: Information on sun exposure has been added: "(>20-30 min/day around moon)

Table 3 – If the study involved sun exposure treatment, briefly state details concerning the sun exposure such as length of time.

Answer: Details regarding length of sun exposure have been added to Table 3.

Line 403 – “complete and accurate”

Answer: It has been corrected.

Major comments

  • The Abstract should include mention of some of the general findings (i.e. Results) found in the review

Answer: The abstract has been corrected according to the suggestion.

  • The authors must be more conservative in their description of the results. For example, on lines 200-201, it is stated that “insulin resistance syndrome … and vitamin D deficiency contribute greatly…cancer.” It would be best to state the “evidence suggests” or it “may contribute”. This is just one example of numerous instances within the manuscript. In the revisions, please specify the line #’s where these corrections were made

Answer: The manuscript has been corrected according to the suggestions:

lines 241, 242, 270, 280, 281, 389, 

  • In-text citations should be included at first mention of a fact. Throughout the manuscript, many facts are stated without a reference.

Answer: Appropriate references have been added.

Round 2

Reviewer 2 Report

The authors have sufficiently addressed all of my comments.